# Learning Sparse Neural Networks through $L_0$ Regularization

**Christos Louizos**[*]
University of Amsterdam
TNO, Intelligent Imaging
c.louizos@uva.nl

**Max Welling**
University of Amsterdam
CIFAR
m.welling@uva.nl

**Diederik P. Kingma**
OpenAI
dpkingma@openai.com

## Abstract

We propose a practical method for $L_0$ norm regularization for neural networks: pruning the network during training by encouraging weights to become exactly zero. Such regularization is interesting since (1) it can greatly speed up training and inference, and (2) it can improve generalization. AIC and BIC, well-known model selection criteria, are special cases of $L_0$ regularization. However, since the $L_0$ norm of weights is non-differentiable, we cannot incorporate it directly as a regularization term in the objective function. We propose a solution through the inclusion of a collection of non-negative stochastic gates, which collectively determine which weights to set to zero. We show that, somewhat surprisingly, for certain distributions over the gates, the expected $L_0$ regularized objective is differentiable with respect to the distribution parameters. We further propose the *hard concrete* distribution for the gates, which is obtained by "stretching" a binary concrete distribution and then transforming its samples with a hard-sigmoid. The parameters of the distribution over the gates can then be jointly optimized with the original network parameters. As a result our method allows for straightforward and efficient learning of model structures with stochastic gradient descent and allows for conditional computation in a principled way. We perform various experiments to demonstrate the effectiveness of the resulting approach and regularizer.

## 1 Introduction

Deep neural networks are flexible function approximators that have been very successful in a broad range of tasks. They can easily scale to millions of parameters while allowing for tractable optimization with mini-batch stochastic gradient descent (SGD), graphical processing units (GPUs) and parallel computation. Nevertheless they do have drawbacks. Firstly, it has been shown in recent works (Han et al., 2015; Ullrich et al., 2017; Molchanov et al., 2017) that they are greatly overparametrized as they can be pruned significantly without any loss in accuracy; this exhibits unnecessary computation and resources. Secondly, they can easily overfit and even memorize random patterns in the data (Zhang et al., 2016), if not properly regularized. This overfitting can lead to poor generalization in practice.

A way to address both of these issues is by employing model compression and sparsification techniques. By sparsifying the model, we can avoid unnecessary computation and resources, since irrelevant degrees of freedom are pruned away and do not need to be computed. Furthermore, we reduce its complexity, thus penalizing memorization and alleviating overfitting.

A conceptually attractive approach is the $L_0$ norm regularization of (blocks of) parameters; this explicitly penalizes parameters for being different than zero with no further restrictions. However, the combinatorial nature of this problem makes for an intractable optimization for large models.

In this paper we propose a general framework for surrogate $L_0$ regularized objectives. It is realized by smoothing the *expected* $L_0$ regularized objective with continuous distributions in a way that can maintain the *exact* zeros in the parameters while still allowing for efficient gradient based optimization. This is achieved by transforming continuous random variables (r.v.s) with a hard nonlinearity, the

---

[*]Work done while interning at OpenAI.

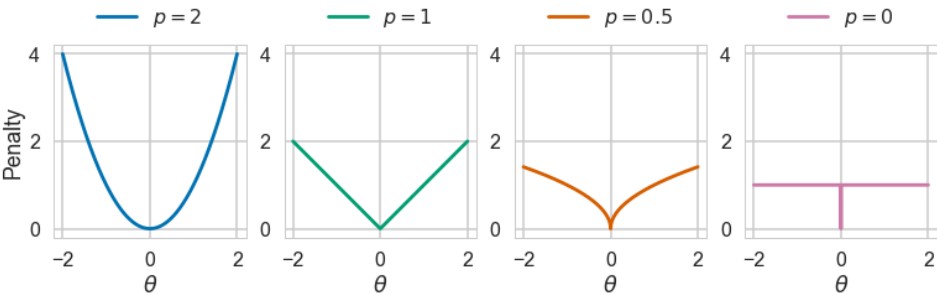

Figure 1: $L_p$ norm penalties for a parameter $\theta$ according to different values of $p$. It is easily observed that both weight decay and Lasso, $p = 2$ and $p = 1$ respectively, impose shrinkage for large values of $\theta$. By gradually allowing $p < 1$ we observe that the shrinkage is reduced and at the limit of $p = 0$ we observe that the penalty is a constant for $\theta \neq 0$.

hard-sigmoid. We further propose and employ a novel distribution obtained by this procedure; the hard concrete. It is obtained by "stretching" a binary concrete random variable (Maddison et al., 2016; Jang et al., 2016) and then passing its samples through a hard-sigmoid. We demonstrate the effectiveness of this simple procedure in various experiments.

## 2  MINIMIZING THE $L_0$ NORM OF PARAMETRIC MODELS

One way to sparsify parametric models, such as deep neural networks, with the least assumptions about the parameters is the following; let $\mathcal{D}$ be a dataset consisting of $N$ i.i.d. input output pairs $\{(\mathbf{x}_1, \mathbf{y}_1), \ldots, (\mathbf{x}_N, \mathbf{y}_N)\}$ and consider a regularized empirical risk minimization procedure with an $L_0$ regularization on the parameters $\boldsymbol{\theta}$ of a hypothesis (e.g. a neural network) $h(\cdot; \boldsymbol{\theta})$[1]:

$$\mathcal{R}(\boldsymbol{\theta}) = \frac{1}{N}\left(\sum_{i=1}^{N}\mathcal{L}\big(h(\mathbf{x}_i; \boldsymbol{\theta}), \mathbf{y}_i\big)\right) + \lambda\|\boldsymbol{\theta}\|_0, \qquad \|\boldsymbol{\theta}\|_0 = \sum_{j=1}^{|\theta|}\mathbb{I}[\theta_j \neq 0], \tag{1}$$

$$\boldsymbol{\theta}^* = \arg\min_{\boldsymbol{\theta}}\{\mathcal{R}(\boldsymbol{\theta})\},$$

where $|\theta|$ is the dimensionality of the parameters, $\lambda$ is a weighting factor for the regularization and $\mathcal{L}(\cdot)$ corresponds to a loss function, e.g. cross-entropy loss for classification or mean-squared error for regression. The $L_0$ norm penalizes the number of non-zero entries of the parameter vector and thus encourages sparsity in the final estimates $\boldsymbol{\theta}^*$. The Akaike Information Criterion (AIC) (Akaike, 1998) and the Bayesian Information Criterion (BIC) (Schwarz et al., 1978), well-known model selection criteria, correspond to specific choices of $\lambda$. Notice that the $L_0$ norm induces no shrinkage on the actual values of the parameters $\boldsymbol{\theta}$; this is in contrast to e.g. $L_1$ regularization and the Lasso (Tibshirani, 1996), where the sparsity is due to shrinking the actual values of $\boldsymbol{\theta}$. We provide a visualization of this effect in Figure 1.

Unfortunately, optimization under this penalty is computationally intractable due to the non-differentiability and combinatorial nature of $2^{|\theta|}$ possible states of the parameter vector $\boldsymbol{\theta}$. How can we relax the discrete nature of the $L_0$ penalty such that we allow for efficient continuous optimization of Eq. 1, while allowing for exact zeros in the parameters? This section will present the necessary details of our approach.

---

[1]This assumption is just for ease of explanation; our proposed framework can be applied to any objective function involving parameters.

## 2.1 A GENERAL RECIPE FOR EFFICIENTLY MINIMIZING $L_0$ NORMS

Consider the $L_0$ norm under a simple re-parametrization of $\boldsymbol{\theta}$:

$$\theta_j = \tilde{\theta}_j z_j, \qquad z_j \in \{0, 1\}, \qquad \tilde{\theta}_j \neq 0, \qquad \|\boldsymbol{\theta}\|_0 = \sum_{j=1}^{|\theta|} z_j, \tag{2}$$

where the $z_j$ correspond to binary "gates" that denote whether a parameter is present and the $L_0$ norm corresponds to the amount of gates being "on". By letting $q(z_j|\pi_j) = \mathrm{Bern}(\pi_j)$ be a Bernoulli distribution over each gate $z_j$ we can reformulate the minimization of Eq. 1 as penalizing the number of parameters being used, on average, as follows:

$$\mathcal{R}(\tilde{\boldsymbol{\theta}}, \boldsymbol{\pi}) = \mathbb{E}_{q(\mathbf{z}|\boldsymbol{\pi})}\left[\frac{1}{N}\left(\sum_{i=1}^{N} \mathcal{L}\big(h(\mathbf{x}_i; \tilde{\boldsymbol{\theta}} \odot \mathbf{z}), \mathbf{y}_i\big)\right)\right] + \lambda \sum_{j=1}^{|\theta|} \pi_j, \tag{3}$$

$$\tilde{\boldsymbol{\theta}}^*, \boldsymbol{\pi}^* = \underset{\tilde{\boldsymbol{\theta}}, \boldsymbol{\pi}}{\arg\min}\{\mathcal{R}(\tilde{\boldsymbol{\theta}}, \boldsymbol{\pi})\},$$

where $\odot$ corresponds to the elementwise product. The objective described in Eq. 3 is in fact a special case of a variational bound over the parameters involving spike and slab (Mitchell & Beauchamp, 1988) priors and approximate posteriors; we refer interested readers to appendix A.

Now the second term of the r.h.s. of Eq. 3 is straightforward to minimize however the first term is problematic for $\boldsymbol{\pi}$ due to the discrete nature of $\mathbf{z}$, which does not allow for efficient gradient based optimization. While in principle a gradient estimator such as the REINFORCE (Williams, 1992) could be employed, it suffers from high variance and control variates (Mnih & Gregor, 2014; Mnih & Rezende, 2016; Tucker et al., 2017), that require auxiliary models or multiple evaluations of the network, have to be employed. Two simpler alternatives would be to use either the straight-through (Bengio et al., 2013) estimator as done at Srinivas et al. (2017) or the concrete distribution as e.g. at Gal et al. (2017). Unfortunately both of these approach have drawbacks; the first one provides biased gradients due to ignoring the Heaviside function in the likelihood during the gradient evaluation whereas the second one does not allow for the gates (and hence parameters) to be exactly zero during optimization, thus precluding the benefits of conditional computation (Bengio et al., 2013).

Fortunately, there is a simple alternative way to smooth the objective such that we allow for efficient gradient based optimization of the expected $L_0$ norm along with zeros in the parameters $\boldsymbol{\theta}$. Let $\mathbf{s}$ be a continuous random variable with a distribution $q(\mathbf{s})$ that has parameters $\boldsymbol{\phi}$. We can now let the gates $\mathbf{z}$ be given by a hard-sigmoid rectification of $\mathbf{s}$[2], as follows:

$$\mathbf{s} \sim q(\mathbf{s}|\boldsymbol{\phi}) \tag{4}$$

$$\mathbf{z} = \min(\mathbf{1}, \max(\mathbf{0}, \mathbf{s})). \tag{5}$$

This would then allow the gate to be *exactly* zero and, due to the underlying continuous random variable $\mathbf{s}$, we can still compute the probability of the gate being non-zero (active). This is easily obtained by the cumulative distribution function (CDF) $Q(\cdot)$ of $\mathbf{s}$:

$$q(\mathbf{z} \neq 0|\boldsymbol{\phi}) = 1 - Q(\mathbf{s} \leq 0|\boldsymbol{\phi}), \tag{6}$$

i.e. it is the probability of the $\mathbf{s}$ variable being positive. We can thus smooth the binary Bernoulli gates $\mathbf{z}$ appearing in Eq. 3 by employing continuous distributions in the aforementioned way:

$$\mathcal{R}(\tilde{\boldsymbol{\theta}}, \boldsymbol{\phi}) = \mathbb{E}_{q(\mathbf{s}|\boldsymbol{\phi})}\left[\frac{1}{N}\left(\sum_{i=1}^{N} \mathcal{L}\big(h(\mathbf{x}_i; \tilde{\boldsymbol{\theta}} \odot g(\mathbf{s})), \mathbf{y}_i\big)\right)\right] + \lambda \sum_{j=1}^{|\theta|} \big(1 - Q(s_j \leq 0|\phi_j)\big), \tag{7}$$

$$\tilde{\boldsymbol{\theta}}^*, \boldsymbol{\phi}* = \underset{\tilde{\boldsymbol{\theta}}, \boldsymbol{\phi}}{\arg\min}\{\mathcal{R}(\tilde{\boldsymbol{\theta}}, \boldsymbol{\phi})\}, \quad g(\cdot) = \min(1, \max(0, \cdot)).$$

Notice that this is a close surrogate to the original objective function in Eq. 3, as we similarly have a cost that explicitly penalizes the probability of a gate being different from zero. Now for continuous

---

[2]We chose to employ a hard-sigmoid instead of a rectifier, $g(\cdot) = \max(0, \cdot)$, so as to have the variable $z$ better mimic a binary gate (rather than a scale variable).

distributions $q(\mathbf{s})$ that allow for the reparameterization trick (Kingma & Welling, 2014; Rezende et al., 2014) we can express the objective in Eq. 7 as an expectation over a parameter free noise distribution $p(\boldsymbol{\epsilon})$ and a deterministic and differentiable transformation $f(\cdot)$ of the parameters $\boldsymbol{\phi}$ and $\boldsymbol{\epsilon}$:

$$\mathcal{R}(\tilde{\boldsymbol{\theta}}, \boldsymbol{\phi}) = \mathbb{E}_{p(\boldsymbol{\epsilon})}\left[\frac{1}{N}\left(\sum_{i=1}^{N}\mathcal{L}\big(h(\mathbf{x}_i; \tilde{\boldsymbol{\theta}} \odot g(f(\boldsymbol{\phi}, \boldsymbol{\epsilon}))), \mathbf{y}_i\big)\right)\right] + \lambda\sum_{j=1}^{|\theta|}\big(1 - Q(s_j \leq 0|\phi_j)\big), \quad (8)$$

which allows us to make the following Monte Carlo approximation to the (generally) intractable expectation over the noise distribution $p(\boldsymbol{\epsilon})$:

$$\hat{\mathcal{R}}(\tilde{\boldsymbol{\theta}}, \boldsymbol{\phi}) = \frac{1}{L}\sum_{l=1}^{L}\left(\frac{1}{N}\left(\sum_{i=1}^{N}\mathcal{L}\big(h(\mathbf{x}_i; \tilde{\boldsymbol{\theta}} \odot \mathbf{z}^{(l)}), \mathbf{y}_i\big)\right)\right) + \lambda\sum_{j=1}^{|\theta|}\big(1 - Q(s_j \leq 0|\phi_j)\big)$$

$$= \mathcal{L}_E(\tilde{\boldsymbol{\theta}}, \boldsymbol{\phi}) + \lambda\mathcal{L}_C(\boldsymbol{\phi}), \quad \text{where } \mathbf{z}^{(l)} = g(f(\boldsymbol{\phi}, \boldsymbol{\epsilon}^{(l)})) \text{ and } \boldsymbol{\epsilon}^{(l)} \sim p(\boldsymbol{\epsilon}). \quad (9)$$

$\mathcal{L}_E$ corresponds to the *error loss* that measures how well the model is fitting the current dataset whereas $\mathcal{L}_C$ refers to the *complexity loss* that measures the flexibility of the model. Crucially, the total cost in Eq. 9 is now differentiable w.r.t. $\boldsymbol{\phi}$, thus enabling for efficient stochastic gradient based optimization, while still allowing for exact zeros at the parameters. One price we pay is that now the gradient of the log-likelihood w.r.t. the parameters $\boldsymbol{\phi}$ of $q(\mathbf{s})$ is sparse due to the rectifications; nevertheless this should not pose an issue considering the prevalence of rectified linear units in neural networks. Furthermore, due to the stochasticity at $\mathbf{s}$ the hard-sigmoid gate $\mathbf{z}$ is smoothed to a soft version on average, thus allowing for gradient based optimization to succeed, even when the mean of $\mathbf{s}$ is negative or larger than one. An example visualization can be seen in Figure 2b. It should be noted that a similar argument was also shown at Bengio et al. (2013), where with logistic noise a rectifier nonlinearity was smoothed to a softplus[3] on average.

## 2.2 THE HARD CONCRETE DISTRIBUTION

The framework described in Section 2.1 gives us the freedom to choose an appropriate smoothing distribution $q(\mathbf{s})$. A choice that seems to work well in practice is the following; assume that we have a binary concrete (Maddison et al., 2016; Jang et al., 2016) random variable $s$ distributed in the $(0, 1)$ interval with probability density $q_s(s|\phi)$ and cumulative density $Q_s(s|\phi)$. The parameters of the distribution are $\phi = (\log\alpha, \beta)$, where $\log\alpha$ is the location and $\beta$ is the temperature. We can "stretch" this distribution to the $(\gamma, \zeta)$ interval, with $\gamma < 0$ and $\zeta > 1$, and then apply a hard-sigmoid on its random samples:

$$u \sim \mathcal{U}(0, 1), \quad s = \text{Sigmoid}\big((\log u - \log(1 - u) + \log\alpha)/\beta\big), \quad \bar{s} = s(\zeta - \gamma) + \gamma, \quad (10)$$

$$z = \min(1, \max(0, \bar{s})). \quad (11)$$

This would then induce a distribution where the probability mass of $q_{\bar{s}}(\bar{s}|\phi)$ on the negative values, $Q_{\bar{s}}(0|\phi)$, is "folded" to a delta peak at zero, the probability mass on values larger than one, $1 - Q_{\bar{s}}(1|\phi)$, is "folded" to a delta peak at one and the original distribution $q_{\bar{s}}(\bar{s}|\phi)$ is truncated to the $(0, 1)$ range. We provide more information and the density of the resulting distribution at the appendix.

Notice that a similar behavior would have been obtained even if we passed samples from any other distribution over the real line through a hard-sigmoid. The only requirement of the approach is that we can evaluate the CDF of $\bar{s}$ at 0 and 1. The main reason for picking the binary concrete is its close ties with Bernoulli r.v.s. It was originally proposed at Maddison et al. (2016); Jang et al. (2016) as a smooth approximation to Bernoulli r.vs, a fact that allows for gradient based optimization of its parameters through the reparametrization trick. The temperature $\beta$ controls the degree of approximation, as with $\beta = 0$ we can recover the original Bernoulli r.v. (but lose the differentiable properties) whereas with $0 < \beta < 1$ we obtain a probability density that concentrates its mass near the endpoints (e.g. as shown in Figure 2a). As a result, the hard concrete also inherits the same theoretical properties w.r.t. the Bernoulli distribution. Furthermore, it can serve as a better approximation of the discrete nature, since it includes $\{0, 1\}$ in its support, while still allowing for (sub)gradient optimization of its parameters due to the continuous probability mass that connects those two values. We can also view this distribution as a "rounded" version of the original binary

---

[3]$f(x) = \log(1 + \exp(x))$.

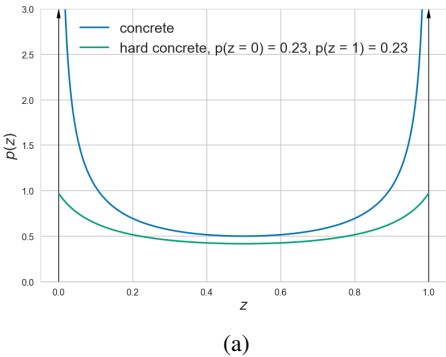
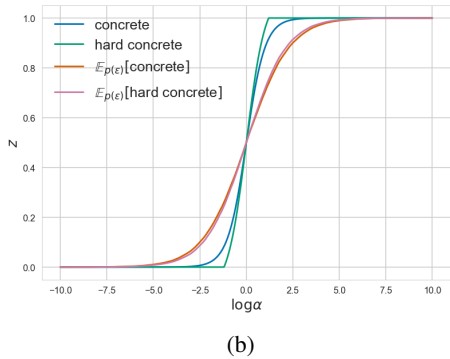

(a)     (b)

Figure 2: **(a)** The binary concrete distribution with location $\log \alpha = 0$ and temperature $\beta = 0.5$ and the hard concrete equivalent distribution obtained by stretching the concrete distribution to $(\gamma = -0.1, \zeta = 1.1)$ and then applying a hard-sigmoid. Under this specification the hard concrete distribution assigns, roughly, half of its mass to $\{0, 1\}$ and the rest to $(0, 1)$. **(b)** The expected value of the afforementioned concrete and hard concrete gate as a function of the location $\log \alpha$, obtained by averaging 10000 samples. We also added the value of the gates obtained by removing the noise entirely. We can see that the noise smooths the hard-sigmoid to a sigmoid on average.

concrete, where values larger than $\frac{1-\gamma}{\zeta - \gamma}$ are rounded to one whereas values smaller than $\frac{-\gamma}{\zeta - \gamma}$ are rounded to zero. We provide an example visualization of the hard concrete distribution in Figure 2a.

The $L_0$ complexity loss of the objective in Eq. 9 under the hard concrete r.v. is conveniently expressed as follows:

$$\mathcal{L}_C = \sum_{j=1}^{|\theta|} \left(1 - Q_{\bar{s}_j}(0|\phi)\right) = \sum_{j=1}^{|\theta|} \text{Sigmoid}\left(\log \alpha_j - \beta \log \frac{-\gamma}{\zeta}\right). \tag{12}$$

At test time we use the following estimator for the final parameters $\theta^*$ under a hard concrete gate:

$$\hat{\mathbf{z}} = \min(\mathbf{1}, \max(\mathbf{0}, \text{Sigmoid}(\log \boldsymbol{\alpha})(\zeta - \gamma) + \gamma)), \qquad \boldsymbol{\theta}^* = \tilde{\boldsymbol{\theta}}^* \odot \hat{\mathbf{z}}. \tag{13}$$

### 2.3 COMBINING THE $L_0$ NORM WITH OTHER NORMS

While the $L_0$ norm leads to sparse estimates without imposing any shrinkage on $\theta$ it might still be desirable to impose some form of prior assumptions on the values of $\theta$ with alternative norms, e.g. impose smoothness with the $L_2$ norm (i.e. weight decay). In the following we will show how this combination is feasible for the $L_2$ norm. The expected $L_2$ norm under the Bernoulli gating mechanism can be conveniently expressed as:

$$\mathbb{E}_{q(\mathbf{z}|\boldsymbol{\pi})}\left[\|\boldsymbol{\theta}\|_2^2\right] = \sum_{j=1}^{|\theta|} \mathbb{E}_{q(z_j|\pi_j)}\left[z_j^2 \tilde{\theta}_j^2\right] = \sum_{j=1}^{|\theta|} \pi_j \tilde{\theta}_j^2, \tag{14}$$

where $\pi_j$ corresponds to the success probability of the Bernoulli gate $z_j$. To maintain a similar expression with our smoothing mechanism, and avoid extra shrinkage for the gates $z_j$, we can take into account that the standard $L_2$ norm penalty is proportional to the negative log density of a zero mean Gaussian prior with a standard deviation of $\sigma = 1$. We will then assume that the $\sigma$ for each $\theta$ is governed by $z$ in a way that when $z = 0$ we have that $\sigma = 1$ and when $z > 0$ we have that $\sigma = z$. As a result, we can obtain the following expression for the $L_2$ penalty (where $\hat{\theta} = \frac{\theta}{\sigma}$):

$$\mathbb{E}_{q(\mathbf{z}|\boldsymbol{\phi})}\left[\|\hat{\boldsymbol{\theta}}\|_2^2\right] = \sum_{j=1}^{|\theta|} \left(Q_{\bar{s}_j}(0|\phi_j)\frac{0}{1} + \left(1 - Q_{\bar{s}_j}(0|\phi_j)\right)\mathbb{E}_{q(z_j|\phi_j,\bar{s}_j>0)}\left[\frac{\tilde{\theta}_j^2 \cancel{z_j^2}}{\cancel{z_j^2}}\right]\right)$$

$$= \sum_{j=1}^{|\theta|} \left(1 - Q_{\bar{s}_j}(0|\phi_j)\right)\tilde{\theta}_j^2. \tag{15}$$

## 2.4 Group sparsity under an $L_0$ norm

For reasons of computational efficiency it is usually desirable to perform group sparsity instead of parameter sparsity, as this can allow for practical computation savings. For example, in neural networks speedups can be obtained by employing a dropout (Srivastava et al., 2014) like procedure with neuron sparsity in fully connected layers or feature map sparsity for convolutional layers (Wen et al., 2016; Louizos et al., 2017; Neklyudov et al., 2017). This is straightforward to do with hard concrete gates; simply share the gate between all of the members of the group. The expected $L_0$ and, according to section 2.3, $L_2$ penalties in this scenario can be rewritten as:

$$\mathbb{E}_{q(\mathbf{z}|\boldsymbol{\phi})}\left[\|\boldsymbol{\theta}\|_0\right] = \sum_{g=1}^{|G|} |g|\left(1 - Q(s_g \leq 0|\phi_g)\right) \tag{16}$$

$$\mathbb{E}_{q(\mathbf{z}|\boldsymbol{\phi})}\left[\|\hat{\boldsymbol{\theta}}\|_2^2\right] = \sum_{g=1}^{|G|} \left((1 - Q(s_g \leq 0|\phi_g)) \sum_{j=1}^{|g|} \tilde{\theta}_j^2\right). \tag{17}$$

where $|G|$ corresponds to the number of groups and $|g|$ corresponds to the number of parameters of group $g$. For all of our subsequent experiments we employed neuron sparsity, where we introduced a gate per input neuron for fully connected layers and a gate per output feature map for convolutional layers. Notice that in the interpretation we adopt the gate is shared across all locations of the feature map for convolutional layers, akin to spatial dropout (Tompson et al., 2015). This can lead to practical computation savings while training, a benefit which is not possible with the commonly used independent dropout masks per spatial location (e.g. as at Zagoruyko & Komodakis (2016)).

## 3 Related work

Compression and sparsification of neural networks has recently gained much traction in the deep learning community. The most common and straightforward technique is parameter / neuron pruning (LeCun et al., 1990) according to some criterion. Whereas weight pruning (Han et al., 2015; Ullrich et al., 2017; Molchanov et al., 2017) is in general inefficient for saving computation time, neuron pruning (Wen et al., 2016; Louizos et al., 2017; Neklyudov et al., 2017) can lead to computation savings. Unfortunately, all of the aforementioned methods require training the original dense network thus precluding the benefits we can obtain by having exact sparsity on the computation during training. This is in contrast to our approach where sparsification happens during training, thus theoretically allowing conditional computation to speed-up training (Bengio et al., 2013; 2015).

Emulating binary r.v.s with rectifications of continuous r.v.s is not a new concept and has been previously done with Gaussian distributions in the context of generative modelling (Hinton & Ghahramani, 1997; Harva & Kabán, 2007; Salimans, 2016) and with logistic distributions at (Bengio et al., 2013) in the context of conditional computation. These distributions can similarly represent the value of exact zero, while still maintaining the tractability of continuous optimization. Nevertheless, they are sub-optimal when we require approximations to binary r.v.s (as is the case for the $L_0$ penalty); we cannot represent the bimodal behavior of a Bernoulli r.v. due to the fact that the underlying distribution is unimodal. Another technique that allows for gradient based optimization of discrete r.v.s are the smoothing transformations proposed by Rolfe (2016). There the core idea is that if a model has binary latent variables, then we can smooth them with continuous noise in a way that allows for reparametrization gradients. There are two main differences with the hard concrete distribution we employ here; firstly, the double rectification of the hard concrete r.v.s allows us to represent the values of exact zero and one (instead of just zero) and, secondly, due to the underlying concrete distribution the random samples from the hard concrete will better emulate binary r.v.s.

## 4 Experiments

We validate the effectiveness of our method on two tasks. The first corresponds to the toy classification task of MNIST using a simple multilayer perceptron (MLP) with two hidden layers of size 300 and 100 (LeCun et al., 1998), and a simple convolutional network, the LeNet-5-Caffe[4]. The second

---

[4]https://github.com/BVLC/caffe/tree/master/examples/mnist

corresponds to the more modern task of CIFAR 10 and CIFAR 100 classification using Wide Residual Networks (Zagoruyko & Komodakis, 2016). For all of our experiments we set $\gamma = -0.1$, $\zeta = 1.1$ and, following the recommendations from Maddison et al. (2016), set $\beta = 2/3$ for the concrete distributions. We initialized the locations $\log \alpha$ by sampling from a normal distribution with a standard deviation of $0.01$ and a mean that yields $\frac{\alpha}{\alpha+1}$ to be approximately equal to the original dropout rate employed at each of the networks. We used a single sample of the gate $\mathbf{z}$ for each minibatch of datapoints during the optimization, even though this can lead to larger variance in the gradients (Kingma et al., 2015). In this way we show that we can obtain the speedups in training with practical implementations, without actually hurting the overall performance of the network.

## 4.1 MNIST CLASSIFICATION AND SPARSIFICATION

For these experiments we did no further regularization besides the $L_0$ norm and optimization was done with Adam (Kingma & Ba, 2014) using the default hyper-parameters and temporal averaging. We can see at Table 1 that our approach is competitive with other methods that tackle neural network compression. However, it is worth noting that all of these approaches prune the network post-training using thresholds while requiring training the full network. We can further see that our approach minimizes the amount of parameters more at layers where the gates affect a larger part of the cost; for the MLP this corresponds to the input layer whereas for the LeNet5 this corresponds to the first fully connected layer. In contrast, the methods with sparsity inducing priors (Louizos et al., 2017; Neklyudov et al., 2017) sparsify parameters irrespective of that extra cost (since they are only encouraged by the prior to move parameters to zero) and as a result they achieve similar sparsity on all of the layers. Nonetheless, it should be mentioned that we can in principle increase the sparsification on specific layers simply by specifying a separate $\lambda$ for each layer, e.g. by increasing the $\lambda$ for gates that affect less parameters. We provide such results at the "$\lambda$ sep." rows.

Table 1: Comparison of the learned architectures and performance of the baselines from Louizos et al. (2017) and the proposed $L_0$ minimization under $L_{0_{hc}}$. We show the amount of neurons left after pruning with the estimator in Eq. 13 along with the error in the test set after 200 epochs. $N$ denotes the number of training datapoints.

| Network & size | Method | Pruned architecture | Error (%) |
|---|---|---|---|
| MLP 784-300-100 | Sparse VD (Molchanov et al., 2017) | 512-114-72 | 1.8 |
| | BC-GNJ (Louizos et al., 2017) | 278-98-13 | 1.8 |
| | BC-GHS (Louizos et al., 2017) | 311-86-14 | 1.8 |
| | $L_{0_{hc}}$, $\lambda = 0.1/N$ | 219-214-100 | 1.4 |
| | $L_{0_{hc}}$, $\lambda$ sep. | 266-88-33 | 1.8 |
| LeNet-5-Caffe 20-50-800-500 | Sparse VD (Molchanov et al., 2017) | 14-19-242-131 | 1.0 |
| | GL (Wen et al., 2016) | 3-12-192-500 | 1.0 |
| | GD (Srinivas & Babu, 2016) | 7-13-208-16 | 1.1 |
| | SBP (Neklyudov et al., 2017) | 3-18-284-283 | 0.9 |
| | BC-GNJ (Louizos et al., 2017) | 8-13-88-13 | 1.0 |
| | BC-GHS (Louizos et al., 2017) | 5-10-76-16 | 1.0 |
| | $L_{0_{hc}}$, $\lambda = 0.1/N$ | 20-25-45-462 | 0.9 |
| | $L_{0_{hc}}$, $\lambda$ sep. | 9-18-65-25 | 1.0 |

To get a better idea about the potential speedup we can obtain in training we plot in Figure 3 the expected, under the probability of the gate being active, floating point operations (FLOPs) as a function of the training iterations. We also included the theoretical speedup we can obtain by using dropout (Srivastava et al., 2014) networks. As we can observe, our $L_0$ minimization procedure that is targeted towards neuron sparsity can potentially yield significant computational benefits compared to the original or dropout architectures, with minimal or no loss in performance. We further observe that there is a significant difference in the flop count for the LeNet model between the $\lambda = 0.1/N$ and $\lambda$ sep. settings. This is because we employed larger values for $\lambda$ ($10/N$ and $0.5/N$) for the convolutional layers (which contribute the most to the computation) in the $\lambda$ sep. setting. As a result,

this setting is more preferable when we are concerned with speedup, rather than network compression (which is affected only by the number of parameters).

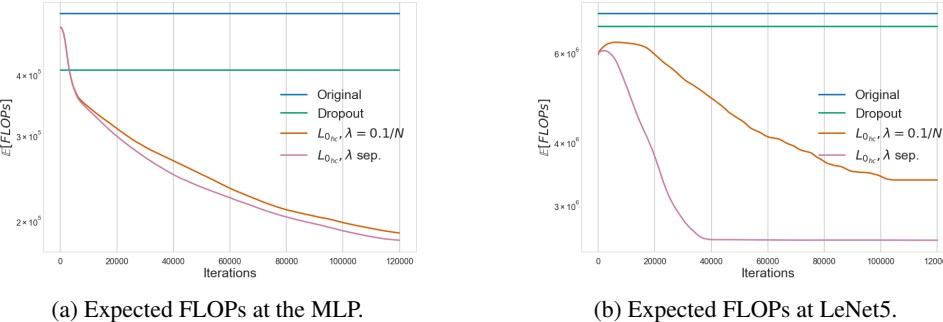

(a) Expected FLOPs at the MLP.       (b) Expected FLOPs at LeNet5.

Figure 3: Expected number of floating point operations (FLOPs) during training for the original, dropout and $L_0$ regularized networks. These were computed by assuming one flop for multiplication and one flop for addition.

## 4.2 CIFAR CLASSIFICATION

For WideResNets we apply $L_0$ regularization on the weights of the hidden layer of the residual blocks, i.e. where dropout is usually employed. We also employed an $L_2$ regularization term as described in Section 2.3 with the weight decay coefficient used in Zagoruyko & Komodakis (2016). For the layers with the hard concrete gates we divided the weight decay coefficient by 0.7 to ensure that a-priori we assume the same length-scale as the 0.3 dropout equivalent network. For optimization we employed the procedure described in Zagoruyko & Komodakis (2016) with a minibatch of 128 datapoints, which was split between two GPUs, and used a single sample for the gates for each GPU.

Table 2: Results on the benchmark classification tasks of CIFAR 10 and CIFAR 100. All of the baseline results are taken from Zagoruyko & Komodakis (2016). For the $L_0$ regularized WRN we report the median of the error on the test set after 200 epochs over 5 runs.

| Network | CIFAR-10 | CIFAR-100 |
|---|---|---|
| original-ResNet-110 (He et al., 2016a) | 6.43 | 25.16 |
| pre-act-ResNet-110 (He et al., 2016b) | 6.37 | - |
| WRN-28-10 (Zagoruyko & Komodakis, 2016) | 4.00 | 21.18 |
| WRN-28-10-dropout (Zagoruyko & Komodakis, 2016) | 3.89 | 18.85 |
| WRN-28-10-$L_{0_{hc}}$, $\lambda = 0.001/N$ | **3.83** | **18.75** |
| WRN-28-10-$L_{0_{hc}}$, $\lambda = 0.002/N$ | 3.93 | 19.04 |

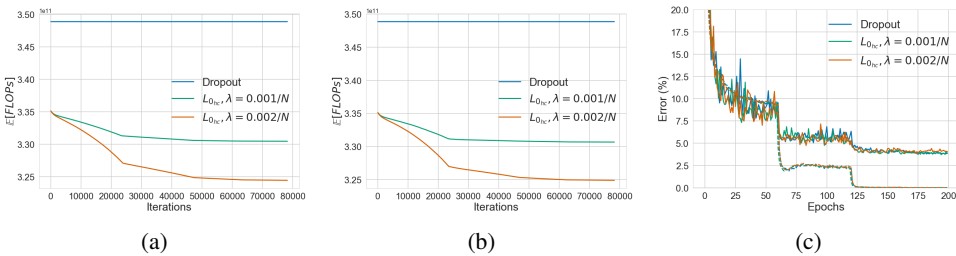

(a)       (b)       (c)

Figure 4: (a, b) Expected number of FLOPs during training for the dropout and $L_0$ regularized WRNs for CIFAR 10 (a) and CIFAR 100 (b). The original WRN is not shown as it has the same practical FLOPs as the dropout equivalent network. (c) Train (dashed) and test (solid) error as a function of the training epochs for dropout and $L_0$ WRNs at CIFAR 10.

As we can observe at Table 2, with a $\lambda$ of $0.001/N$ the $L_0$ regularized wide residual network improves upon the accuracy of the dropout equivalent network on both CIFAR 10 and CIFAR 100. Furthermore, it simultaneously allows for potential training time speedup due to gradually decreasing the number of FLOPs, as we can see in Figures 4a, 4b. This sparsity is also obtained without any "lag" in convergence speed, as at Figure 4c we observe a behaviour that is similar to the dropout network. Finally, we observe that by further increasing $\lambda$ we obtain a model that has a slight error increase but can allow for a larger speedup.

## 5   DISCUSSION

We have described a general recipe that allows for optimizing the $L_0$ norm of parametric models in a principled and effective manner. The method is based on smoothing the combinatorial problem with continuous distributions followed by a hard-sigmoid. To this end, we also proposed a novel distribution which we coin as the hard concrete; it is a "stretched" binary concrete distribution, the samples of which are transformed by a hard-sigmoid. This in turn better mimics the binary nature of Bernoulli distributions while still allowing for efficient gradient based optimization. In experiments we have shown that the proposed $L_0$ minimization process leads to neural network sparsification that is competitive with current approaches while theoretically allowing for speedup in training. We have further shown that this process can provide a good inductive bias and regularizer, as on the CIFAR experiments with wide residual networks we improved upon dropout.

As for future work; better harnessing the power of conditional computation for efficiently training very large neural networks with learned sparsity patterns is a potential research direction. It would be also interesting to adopt a full Bayesian treatment over the parameters $\boldsymbol{\theta}$, such as the one employed at Molchanov et al. (2017); Louizos et al. (2017). This would then allow for further speedup and compression due to the ability of automatically learning the bit precision of each weight. Finally, it would be interesting to explore the behavior of hard concrete r.v.s at binary latent variable models, since they can be used as a drop in replacement that allow us to maintain both the discrete nature as well as the efficient reparametrization gradient optimization.

## ACKNOWLEDGEMENTS

We would like to thank Taco Cohen, Thomas Kipf, Patrick Forré, and Rianne van den Berg for feedback on an early draft of this paper.

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

APPENDIX

## A    RELATION TO VARIATIONAL INFERENCE

The objective function described in Eq. 3 is in fact a special case of a variational lower bound over the parameters of the network under a spike and slab (Mitchell & Beauchamp, 1988) prior. The spike and slab distribution is the golden standard in sparsity as far as Bayesian inference is concerned and it is defined as a mixture of a delta spike at zero and a continuous distribution over the real line (e.g. a standard normal):

$$p(z) = \text{Bernoulli}(\pi), \qquad p(\theta|z=0) = \delta(\theta), \qquad p(\theta|z=1) = \mathcal{N}(\theta|0,1). \qquad (18)$$

Since the true posterior distribution over the parameters under this prior is intractable, we will use variational inference (Beal, 2003). Let $q(\theta, z)$ be a spike and slab approximate posterior over the parameters $\theta$ and gate variables $z$, where we assume that it factorizes over the dimensionality of the parameters $\theta$. It turns out that we can write the following variational free energy under the spike and

slab prior and approximate posterior over a parameter vector $\boldsymbol{\theta}$:

$$\mathcal{F} = -\mathbb{E}_{q(\mathbf{z})q(\boldsymbol{\theta}|\mathbf{z})}[\log p(\mathcal{D}|\boldsymbol{\theta})] + \sum_{j=1}^{|\theta|} KL(q(z_j)||p(z_j)) +$$

$$+ \sum_{j=1}^{|\theta|} \big( q(z_j = 1) KL(q(\theta_j|z_j = 1)||p(\theta_j|z_j = 1)) +$$

$$+ q(z_j = 0) KL(q(\theta_j|z_j = 0)||p(\theta_j|z_j = 0)) \big) \tag{19}$$

$$= -\mathbb{E}_{q(\mathbf{z})q(\boldsymbol{\theta}|\mathbf{z})}[\log p(\mathcal{D}|\boldsymbol{\theta})] + \sum_{j=1}^{|\theta|} KL(q(z_j)||p(z_j)) +$$

$$+ \sum_{j=1}^{|\theta|} q(z_j = 1) KL(q(\theta_j|z_j = 1)||p(\theta_j|z_j = 1)), \tag{20}$$

where the last step is due to $KL(q(\theta_j|z_j = 0)||p(\theta_j|z_j = 0)) = 0$[5]. The term that involves $KL(q(z_j)||p(z_j))$ corresponds to the KL-divergence from the Bernoulli prior $p(z_j)$ to the Bernoulli approximate posterior $q(z_j)$ and $KL(q(\theta_j|z_j = 1)||p(\theta_j|z_j = 1))$ can be interpreted as the "code cost" or else the amount of information the parameter $\theta_j$ contains about the data $\mathcal{D}$, measured by the KL-divergence from the prior $p(\theta_j|z_j = 1)$.

Now consider making the assumption that we are optimizing, rather than integrating, over $\boldsymbol{\theta}$ and further assuming that $KL(q(\theta_j|z_j = 1)||p(\theta_j|z_j = 1)) = \lambda$. We can justify this assumption from an empirical Bayesian procedure: there is a hypothetical prior for each parameter $p(\theta_j|z_j = 1)$ that adapts to $q(\theta_j|z_j = 1)$ in a way that results into needing, approximately, $\lambda$ nats to transform $p(\theta_j|z_j = 1)$ to that particular $q(\theta_j|z_j = 1)$. Those $\lambda$ nats are thus the amount of information the $q(\theta_j|z_j = 1)$ can encode about the data had we used that $p(\theta_j|z_j = 1)$ as the prior. Notice that under this view we can consider $\lambda$ as the amount of flexibility of that hypothetical prior; with $\lambda = 0$ we have a prior that is flexible enough to represent exactly $q(\theta_j|z_j = 1)$, thus resulting into no code cost and possible overfitting. Under this assumption the variational free energy can be re-written as:

$$\mathcal{F} = -\mathbb{E}_{q(\mathbf{z})}[\log p(\mathcal{D}|\tilde{\boldsymbol{\theta}} \odot \mathbf{z})] + \sum_{j=1}^{|\theta|} KL(q(z_j)||p(z_j)) + \lambda \sum_{j=1}^{|\theta|} q(z_j = 1) \tag{21}$$

$$\geq -\mathbb{E}_{q(\mathbf{z})}[\log p(\mathcal{D}|\tilde{\boldsymbol{\theta}} \odot \mathbf{z})] + \lambda \sum_{j=1}^{|\theta|} \pi_j, \tag{22}$$

where $\tilde{\boldsymbol{\theta}}$ corresponds to the optimized $\boldsymbol{\theta}$ and the last step is due to the positivity of the KL-divergence. Now by taking the negative log-probability of the data to be equal to the loss $\mathcal{L}(\cdot)$ of Eq. 1 we see that Eq. 22 is the same as Eq. 3. Note that in case that we are interested over the uncertainty of the gates $\mathbf{z}$, we should optimize Eq. 21, rather than Eq. 22, as this will properly penalize the entropy of $q(\mathbf{z})$. Furthermore, Eq. 21 also allows for the incorporation of prior information about the behavior of the gates (e.g. gates being active 10% of the time, on average). We have thus shown that the expected $L_0$ minimization procedure is in fact a close surrogate to a variational bound involving a spike and slab distribution over the parameters and a fixed coding cost for the parameters when the gates are active.

## B   THE HARD CONCRETE DISTRIBUTION

As mentioned in the main text, the hard concrete is a straightforward modification of the binary concrete (Maddison et al., 2016; Jang et al., 2016); let $q_s(s|\phi)$ be the probability density function

---

[5]We can see that this is indeed the case by taking the limit of $\sigma \to 0$ of the KL divergence of two Gaussians that have the same mean and variance.

(pdf) and $Q_s(s|\phi)$ the cumulative distribution function (CDF) of a binary concrete random variable $s$:

$$q_s(s|\phi) = \frac{\beta\alpha s^{-\beta-1}(1-s)^{-\beta-1}}{(\alpha s^{-\beta} + (1-s)^{-\beta})^2}, \tag{23}$$

$$Q_s(s|\phi) = \text{Sigmoid}((\log s - \log(1-s))\beta - \log\alpha). \tag{24}$$

Now by stretching this distribution to the $(\gamma, \zeta)$ interval, with $\gamma < 0$ and $\zeta > 1$ we obtain $\bar{s} = s(\zeta - \gamma) + \gamma$ with the following pdf and CDF:

$$q_{\bar{s}}(\bar{s}|\phi) = \frac{1}{|\zeta - \gamma|} q_s\left(\frac{\bar{s} - \gamma}{\zeta - \gamma}\Big|\phi\right), \qquad Q_{\bar{s}}(\bar{s}|\phi) = Q_s\left(\frac{\bar{s} - \gamma}{\zeta - \gamma}\Big|\phi\right). \tag{25}$$

and by further rectifying $\bar{s}$ with the hard-sigmoid, $z = \min(1, \max(0, \bar{s}))$, we obtain the following distribution over $z$:

$$q(z|\phi) = Q_{\bar{s}}(0|\phi)\delta(z) + (1 - Q_{\bar{s}}(1|\phi))\delta(z-1) + (Q_{\bar{s}}(1|\phi) - Q_{\bar{s}}(0|\phi))q_{\bar{s}}(z|\bar{s} \in (0,1), \phi), \tag{26}$$

which is composed by a delta peak at zero with probability $Q_{\bar{s}}(0|\phi)$, a delta peak at one with probability $1 - Q_{\bar{s}}(1|\phi)$, and a truncated version of $q_{\bar{s}}(\bar{s}|\phi)$ in the (0, 1) range.

## C   NEGATIVE KL-DIVERGENCE FOR HARD CONCRETE DISTRIBUTIONS

In case th 21 is to be optimized with a hard concrete $q(z)$ then we have to compute the KL-divergence from a prior $p(z)$ to $q(z)$. It is necessary for the prior $p(z)$ to have the same support as $q(z)$ in order for the KL-divergence to be valid; as a result we can let the prior $p(z)$ similarly be a hard-sigmoid transformation of an arbitrary continuous distribution $p(\bar{s})$ with CDF $P_{\bar{s}}(\bar{s})$:

$$p(z) = P_{\bar{s}}(0)\delta(z) + (1 - P_{\bar{s}}(1))\delta(z-1) + (P_{\bar{s}}(1) - P_{\bar{s}}(0))p_{\bar{s}}(z|\bar{s} \in (0,1)) \tag{27}$$

Since both $q(z)$ and $p(z)$ are mixtures with the same number of components we can use the chain rule of relative entropy (Cover & Thomas, 2012; Hershey & Olsen, 2007) in order to compute the KL-divergence:

$$KL(q(z)||p(z)) = Q_{\bar{s}}(0)\log\frac{Q_{\bar{s}}(0)}{P_{\bar{s}}(0)} + (1 - Q_{\bar{s}}(1))\log\frac{1 - Q_{\bar{s}}(1)}{1 - P_{\bar{s}}(1)} +$$
$$+ (Q_{\bar{s}}(1) - Q_{\bar{s}}(0))\,\mathbb{E}_{q_{\bar{s}}(z|\bar{s}\in(0,1))}[\log q_{\bar{s}}(z) - \log p_{\bar{s}}(z)], \tag{28}$$

where $\bar{s}$ corresponds to the the pre-rectified variable. Notice that in case that the integral under the truncated distribution $q(\bar{s}|\bar{s} \in (0,1))$ is not available in closed form we can still obtain a Monte Carlo estimate by sampling the truncated distribution, on e.g. a $(\gamma, \zeta)$ interval, via the inverse transform method:

$$u \sim \mathcal{U}(0,1), \qquad z = Q_{\bar{s}}^{-1}\big(Q_{\bar{s}}(\gamma) + u(Q_{\bar{s}}(\zeta) - Q_{\bar{s}}(\gamma))\big), \tag{29}$$

where $Q_{\bar{s}}^{-1}(\cdot)$ corresponds to the quantile function and $Q_{\bar{s}}(\cdot)$ to the CDF of the random variable $\bar{s}$. Furthermore, it should be mentioned that $KL(q(z)||p(z)) \neq KL(q(\bar{s})||p(\bar{s}))$, since the rectifications are not invertible transformations.

