# OpenReview forum: "Learning Sparse Neural Networks through L_0 Regularization"
_ICLR.cc/2018/Conference — Accept (Poster)_

### Official Review · AnonReviewer3 · 2017-11-25
**This paper constructs a continuous surrogate for ell_0 norm via simple modifications of the binary concrete relaxation with additional stretching and hard-sigmoid transformation steps. The method is easily implemented and empirically seems effective.**

**Rating:** 6
**Confidence:** 3

**Review:**

This paper presents a continuous surrogate for the ell_0 norm and focuses on its applications in regularized empirical regularized minimization. The proposed continuous relaxation scheme allows for gradient based-stochastic optimization for binary discrete variables under the reparameterization trick, and extends the original binary concrete distribution by allowing the parameter taking values of exact zeros and ones, with additional stretching and thresholding operations. Under a compound construction of sparsity, the proposed approach can easily incorporate group sparsity by sharing supports among the grouped variables, or be combined with other types of regularizations on the magnitude of non-zero components. The efficacy of the proposed method in sparsification and speedup is demonstrated in two experiments with comparisons against a few baseline methods.

Pros:

- The paper is clearly written, self-contained and a pleasure to read.
- Based on the evidence provided, the procedure seems to be a useful continuous relaxation scheme to consider in handling optimization with spike and slab regularization

Cons:

- It would be interesting to see how the induced penalty behaves in terms shrinkage comparing against ell_0 and other ell_p choices
- It is unclear what properties does the proposed hard-concrete distribution have, e.g., closed-form density, convexity, etc.
- If the authors can offer a rigorous analysis on the influence of base concrete distribution and provide more guidance on how to choose the stretching parameters in practice, this paper would be more significant

---

> ### Author Response · Authors · 2018-01-02
> **Included a comparison against an L1 regularization baseline, more information about the hard concrete can be found in the appendix and the stretching parameters are not very important.**
>
> We would first like to thank you for taking the time to review our submission; we will now address your comments:
>
> - We agree that comparing other L_p choices with the L_0 norm is beneficial; we would like to point out that the GL (Group Lasso) baseline method for the LeNet5-Caffe experiment employs L_1 regularization for pruning neurons and convolutional filters so we believe that it can serve as a way to measure differences between the L_0 norm and the most popular L_p alternative.
>
> - Due to lack of space we provided a bit of more information about the hard concrete distribution at the appendix; it has a closed-form density that involves the CDF and PDF of the concrete distribution.
>
> - The stretching parameters were initially chosen heuristically and kept fixed for all of the experiments. The heuristic was to approximately aim for clipping to zero if the value of the random variable is less than 0.1 or rounding to 1 if the value of the random variable is larger than 0.9. It should be noted that their choice is not particularly important due to their interplay with the temperature parameter of the concrete distribution; they collectively determine the probabilities of the endpoints {0, 1}, i.e. p(z=0) and p(z=1). As a result we believe that the choice of the stretching parameters is not very important, given the fact that the temperature of the concrete distribution is tuned appropriately.

---

### Official Review · AnonReviewer2 · 2017-11-27
**A sensible and interesting approach to parameter estimation with L0 sparsity which could potentially be applied to many different models. However, in this specific application to standard feedforward network the main advantage was a theoretical speed-up and a robust empirical demonstration over other methods was missing.**

**Rating:** 6
**Confidence:** 3

**Review:**

Learning sparse neural networks through L0 regularisation

Summary:

The authors introduce a gradient-based approach to minimise an objective function with an L0 sparse penalty. The problem is relaxed onto a continuous optimisation by changing an expectation over discrete variables (representing whether a variable is present or not) to an expectation over continuous variables, inspired by earlier work from Maddison et al (ICLR 2017) where a similar transformation was used to learn over discrete variable prediction tasks with neural networks. Here the application is to learn sparse feedforward networks in standard classification tasks, although the framework described is quite general and could be used to impose L0 sparsity to any objective function in principal. The method provides equivalent accuracy and sparsity to published state-of-the-art results on these datasets but it is argue that learning sparsity during the training process will lead to significant speed-ups - this is demonstrated by comparing to a theoretical benchmark (standard training with dropout) rather than through empirical testing against other implementations.

Pros:

The paper is well written and the derivation of the method is easy to follow with a good explanation of the underlying theory.

Optimisation under L0 regularisation is a difficult and generally important topic and certainly has advantages over other sparse inference objective functions that impose shrinkage on non-sparse parameters.

The work is put in context and related to some previous relaxation approaches to sparsity.

The method allows for sparsity to be learned during training rather than after training (as in standard dropout approaches) and this allows the algorithm to obtain significant per-iteration speed-ups, which improves through training.

Cons:

The method is applied to standard neural network architectures and performance in terms of accuracy and final achieved sparsity is comparable to the state-of-the-art methods. Therefore the main advance is in terms of learning speed to obtain this similar performance. However, the learning speed-up is presented against a theoretical FLOPs estimate per iteration for a similar network with dropout. It would be useful to know whether the number of iterations to achieve a particular performance is equivalent for all the different architectures considered, e.g. does the proposed sparse learning method converge at the same rate as the others? I felt a more thorough experimental section would have greatly improved the work, focussing on this learning speed aspect.

It was unclear how much tuning of the lambda hyper-parameter, which tunes the sparsity, would be required in a practical application since tuning this parameter would increase computation time. It might be useful to provide a full Bayesian treatment so that the optimal sparsity can be chosen through hyper-parameter learning.

Minor point: it wasn’t completely clear to me why the fact (3) is a variational approximation to a spike-and-slab is important (Appendix). I don’t see why the spike-and-slab is any more fundamental than the L0 norm prior in (2), it is just more convenient in Bayesian inference because it is an iid prior and potentially allows an informative prior over each parameter. In the context here this didn’t seem a particularly relevant addition to the paper.

---

> ### Author Response · Authors · 2018-01-02
> **Convergence speed is similar to the dropout equivalent networks, lambda hyperparameter didn't need a lot of tuning and the spike and slab connection can allow for the incorporation of prior information in the sparsity mechanism.**
>
> We would first like to thank you for the thorough and extensive review. Regarding whether the method converges in a similar way to standard networks; indeed this is the case. On the CIFAR task with WRNs the L0 regularized networks had similar learning curves with the dropout equivalent networks. We have updated the paper with an example plot on CIFAR 10.
>
> Regarding the lambda hyperparameter; this is true. Empirically, we didn’t have to tune this parameter a lot and considered a small set of values. Treating this parameter in a Bayesian way would indeed be a fruitful direction for future research.
>
> As for the spike-and-slab connection; we agree that it is a minor point (hence it is in the appendix), but we still believe that it is a relevant addition to the paper. It provides an interpretation to the L0 objective that also allows for the incorporation of prior knowledge about the behaviour of the sparity in the form of a prior over the gates. This could then potentially allow for better regularization of the gating mechanism.

---

### Official Review · AnonReviewer1 · 2017-11-28
**Important problem with convincing results but lack some important comparisons**

**Rating:** 7
**Confidence:** 4

**Review:**

The paper introduces a technique for optimizing an L0 penalty on the weights of a neural network. The basic problem is empirical risk minimization with a incremental penalty for each non zero weight. To tackle this problem, this paper proposes an expected surrogate loss that is then relaxed using a method related to recently introduced relaxations of discrete random variables. The authors note that this loss can also be seen as a specific variational bound of a Bayesian model over the weights. The key advantage of this method is that it gives a training time technique for sparsifying neural network computation, leading to potential wins in computation time during training.

The results presented in the paper are convincing. They achieve results competitive with previous methods, with the additional advantage that their sparse models are available during training time. They show order of magnitude reductions in computation time for small models, and more modest constant improvements for large models. The hard concrete distribution is a small but nice contribution on its own.

My only concern is the lack of discussion on the relationship between this method and Concrete Dropout (https://arxiv.org/abs/1705.07832). Although the focus is apparently different, these methods are clearly closely related. A discussion of this relationship seems really important.

Specific comments/questions:
- The reduction of computation time is the key advantage, and it would have been nice to see a more thorough investigation of this. For example, it would have been interesting to see whether this method would work with structured L0 penalties that removed entire units (as opposed to single weights) or other subsets of the computation. This would give a stronger sense of the kind of wins that are possible in this framework.
- Hard concrete is a nice contribution, but there are clearly many possibilities for these relaxations. Extra evaluations of different relaxations would be appreciated. At the very least a comparison to concrete would be nice.
- In equation 2, the equality of the L0 norm with the sum of z assumes that tilde{theta} is not 0.

---

> ### Author Response · Authors · 2018-01-02
> **Included discussion about concrete dropout in the paper; it does not allow for values of exact zero thus it cannot accommodate for conditional computation (which was one of the main objectives of this work).**
>
> We would first like to thank you for the constructive review; we revised the paper and now it contains a discussion of concrete Dropout. The main difference is that concrete dropout does not allow for values of exact zero (and one) thus precluding the benefits of sparsity during training time. One potential way to employ concrete Dropout in this case would be to use it as a biased surrogate for the optimization of eq. 3; this could still allow for potential sparsity at test time by pruning according to thresholds, but nevertheless would require evaluating the full original model during training. As for your other comments:
>
> - Perhaps it's not very prominent but all of our results employ structured penalties, i.e. we are removing either entire convolutional feature maps or entire hidden units.
>
> - For the reasons we previously mentioned, we believe that comparing with concrete dropout will not provide much extra information as the sparsity could only be achieved at test time and not during training (which was one of the main objectives of this work). An alternative that maintains the sparsity during training time, and we experimented with in a pilot study, was a smoothing mechanism that involved the hard-sigmoid of a Gaussian r.v.. This turned out to be worse than the hard concrete procedure, and we attribute this to the unimodality of the underlying Gaussian distribution (which cannot accurately capture the behaviour of a Bernoulli r.v.). We mentioned a couple of sentences about this in the related work section. We also included a comparison against “Generalized Dropout” (GD) that utilizes the straight-through estimator for the same LeNet-5 task we considered in the experiments; this can serve as a comparison against the proposed hard concrete smoothing procedure.
>
> - This is indeed true and we have updated the text accordingly.

---

### Public Comment · ~Suraj_Srinivas1 · 2017-12-09
**Large overlap with previous work**

Hello all,

This paper has large overlap with my own work which was published previously.
Paper 1 - "Learning Sparse Neural Networks" https://arxiv.org/abs/1611.06694  (published in 2017 CVPR workshop)

The section on group sparsity is also very similar to my earlier work.
Paper 2 - "Learning Neural Network Architectures using Backpropagation" https://arxiv.org/abs/1511.05497 (published in BMVC 2016)

Similarities:
1) We motivate the problem as an intractable L0 regularization problem, which is equation 1 in both of those papers (although we do not use the term L0 to describe it).
2) We propose using binary gates for every weight and a regularizer which sums over 'smoothened' values of gates.
3) We make a connection to spike-and-slab priors (in Paper 1).
4) We view the process as an intractable monte-carlo sum (in Paper 1), but we additionally introduce a variance-reducing term.

Differences:
1) We use a bernoulli distribution directly with a straight-through estimator[1] in the optimization whereas this paper uses a concrete distribution.
2) We have an additional regularization term to make learning with bernoulli distributions stable (i.e.; variance reducing term).

We believe that both our papers (Paper 1 and 2) and this paper attempt to solve the same overall objective, but use slightly different relaxation methods.

Thanks,
Suraj Srinivas

[1]: Yoshua Bengio, Nicholas Léonard, and Aaron Courville. Estimating or propagating gradients through
stochastic neurons for conditional computation. arXiv preprint arXiv:1308.3432, 2013.

---

> ### Author Response · Authors · 2018-01-02
> **The overlap is relatively small as the similarities end at eq. 3.**
>
> We would like to thank you for taking an interest in our work and pointing out yours, we were not aware of it. After reading the relevant papers we agree that there indeed are some similarities (but also a large amount of differences) and we have updated our submission accordingly. More specifically, we believe that the similarities between our works end at eq. 3 which provides the expected L0 regularized objective under a Bernoulli gating mechanism; your paper 1 proceeded in optimizing that objective with the biased straight through estimator for the gradients of the discrete gates. This is also what we mentioned as an alternative at the paragraph underneath eq. 3. Notice that this objective is not differentiable w.r.t. the parameters of the gates as you have to take the gradient of the Heaviside function. Our main contribution is to show how we can smooth the L0 regularized objective in a way that can make it differentiable, and thus allow for efficient gradient based optimization, without needing extra terms to make learning stable. The hard concrete distribution was then one potential instance of that framework, but certainly not the only choice.

---

### Decision · Program_Chairs · 2018-01-29
**ICLR 2018 Conference Acceptance Decision**

**Decision:**

Accept (Poster)

**Comment:**

The results in the paper are interesting, and the modifications improve the paper further. Reviewers found teh paper interesting and potentailly applicable to many models.